# Dietary Se-Enriched *Cardamine enshiensis* Supplementation Alleviates Transport-Stress-Induced Body Weight Loss, Anti-Oxidative Capacity and Meat Quality Impairments of Broilers

**DOI:** 10.3390/ani12223193

**Published:** 2022-11-18

**Authors:** Xiao Xu, Yunfen Zhu, Yu Wei, Xiaofei Chen, Rong Li, Junhui Xie, Guogui Wang, Jiajia Ming, Hongqing Yin, Jiqian Xiang, Feiruo Huang, Yongkang Yang

**Affiliations:** 1Enshi Tujia & Miao Autonomous Prefecture Academy of Agricultural Science, Enshi 445000, China; 2College of Animal Science and Technology, Huazhong Agricultural University, Wuhan 430070, China; 3Key Laboratory of Animal Nutrition and Feed Science, Wuhan Polytechnic University, Wuhan 430023, China

**Keywords:** anti-oxidative capacity, broilers, *Cardamine enshiensis*, meat quality, selenium, transport stress

## Abstract

**Simple Summary:**

Meat quality is an important factor influencing the desire of people to purchase chicken meat, and ultimately their satisfaction. However, the meat quality of chicken is often affected by pre-slaughter transportation. Transport stress changes the metabolism as well as psychological status of broilers, which leads to production of extensive reactive oxygen species. Selenium (Se) plays an important role in alleviating oxidative stress of livestock and the most common Se source is sodium selenite (Na_2_SeO_3_) as an inorganic form. Se-enriched *Cardamine enshiensis* (SeCe) is a newly discovered *Brassicaceae* plant which is known to hyperaccumulate Se, and some studies have showed the positive effects of SeCe on performance and anti-oxidative capacity in humans and animals. However, there is very little evidence reporting the effects of dietary SeCe on performance and meat quality of broilers under transport stress. Therefore, the objective of this study was to compare the effects of two sources of Se (Na_2_SeO_3_ vs. SeCe) on body weight loss, anti-oxidative capacity and meat quality of broilers under transport stress. The results showed that a supplement of 0.3 mg/kg of Se sourced from SeCe alleviated transport stress-induced body weight loss, anti-oxidative capacity and meat quality impairments of broilers compared with the broilers fed 0.3 mg/kg of Se sourced from Na_2_SeO_3_ diets.

**Abstract:**

The aim of this experiment was to explore the effects of a new selenium (Se) source from Se-enriched *Cardamine enshiensis* (SeCe) on body weight loss, anti-oxidative capacity and meat quality of broilers under transport stress. A total of 240 one-day-old ROSS 308 broilers were allotted into four treatments with six replicate cages and 10 birds per cage using a 2 × 2 factorial design. The four groups were as follows: (1) Na_2_SeO_3_-NTS group, dietary 0.3 mg/kg Se from Na_2_SeO_3_ without transport stress, (2) SeCe-NTS group, dietary 0.3 mg/kg Se from SeCe without transport stress, (3) Na_2_SeO_3_-TS group, dietary 0.3 mg/kg Se from Na_2_SeO_3_ with transport stress, and (4) SeCe-TS group, dietary 0.3 mg/kg Se from SeCe with transport stress. After a 42 d feeding period, the broilers were transported by a lorry or kept in the original cages for 3 h, respectively. The results showed that dietary SeCe supplementation alleviated transport-stress-induced body weight loss and hepatomegaly of the broilers compared with the broilers fed Na_2_SeO_3_ diets (*p* < 0.05). Furthermore, dietary SeCe supplementation increased the concentrations of plasma total protein and glucose, and decreased the activities of aspartate aminotransferase and alanine aminotransferase of the broilers under transport stress (*p* < 0.05). Dietary SeCe supplementation also enhanced the anti-oxidative capacity and meat quality in the breast and thigh muscles of the broilers under transport stress (*p* < 0.05). In summary, compared with Na_2_SeO_3_, dietary SeCe supplementation alleviates transport-stress-induced body weight loss, anti-oxidative capacity and meat quality impairments of broilers.

## 1. Introduction

In the modern poultry industry, market-age broilers are transported to slaughterhouses. Transport stress is a complex stress which is usually combined with several stressors such as catching, uploading and offloading of broilers into vehicles without feed and water [1]. Transport stress has been reported to induce various physiological and metabolic changes detrimental to animal welfare, health status and meat quality, leading to great economic loss to animal production [2,3]. Transport stress can increase the reactive oxygen species (ROS) level which usually causes body weight loss, anti-oxidative capacity reduction and adverse effects on meat flavor, color and nutritional value [4]. Therefore, the poultry industry is concerned about how to alleviate transport stress. Dietary supplementation with certain nutrients to mitigate transport stress and improve meat quality of transported broilers has been reported [2,5].

Selenium (Se) is a necessary trace element for poultry and livestock. Se plays an important role in alleviating oxidative stress and inflammation because it is a cofactor of many anti-oxidases such as glutathione peroxidase (GSH-Px) [6]. Generally, Se is supplemented in the diets as inorganic forms such as sodium selenite (Na_2_SeO_3_), and organic forms such as Se-enriched yeast [7,8]. *Cardamine enshiensis*, also known as *Cardamine violifolia*, is a newly discovered *Brassicaceae* plant which is known to hyperaccumulate Se [9]. The concentration of Se in *C. enshiensis* was reported to reach 4.4 g/kg dry matter, and the major form of Se was occurs as selenomethionine (SeMet) [10]. Studies have shown that Se-enriched *C. enshiensis* (SeCe) modulates oxidative stress and gut microbiota in _D_-galactose-injected mice [11]. Our previous studies found that SeCe improved growth performance and meat quality in broilers compared with Na_2_SeO_3_ and Se-enriched yeast [12]. Additionally, SeCe had better effects on enhancing anti-oxidative capacity and meat quality of LPS-injected broilers compared with Na_2_SeO_3_ [13].

However, there has been little work reported on the effects of SeCe on performance and meat quality in transport-stressed broilers. Based on previous studies, we hypothesized that the addition of SeCe in broiler’s diets could alleviate performance and meat quality impairments induced by transport stress. Accordingly, the objective of this study was to compare two sources (Na_2_SeO_3_ vs. SeCe) of Se on the performance and meat quality in broilers under transport stress.

## 2. Materials and Methods

### 2.1. Experimental Selenium Sources

Experimental Se sources of Na_2_SeO_3_ and SeCe were provided by Enshi Tujia and Miao Autonomous Prefecture Academy of Agricultural Sciences (Enshi, Hubei, China). The purity of Na_2_SeO_3_ exceeded 99.9%. The Se content in the SeCe powder was 1400 mg/kg dry matter. The forms of Se in SeCe mainly included SeMet, selenocysteine (SeCys_2_), and methylselenocysteine (MeSeCys).

### 2.2. Experimental Birds, Diets and Design

The protocol (No. ETMAPAAS20220409) conducted in this study was approved by the Institutional Animal Care and Use Committee of Enshi Tujia & Miao Autonomous Prefecture Academy of Agricultural Science (Enshi, China). A total of 240 one-day-old male ROSS 308 broilers were purchased from the Xiangyang Charoen Pokphand Co., Ltd., Xiangyang, Hubei, China. Birds were randomly divided into four treatments with six replicates per treatment and 10 birds per replicate in a 2 × 2 factorial design. The four groups were as follows: (1) Na_2_SeO_3_-NTS group, dietary 0.3 mg/kg Se from Na_2_SeO_3_ without transport stress, (2) SeCe-NTS group, dietary 0.3 mg/kg Se from SeCe without transport stress, (3) Na_2_SeO_3_-TS group, dietary 0.3 mg/kg Se from Na_2_SeO_3_ with transport stress, and (4) SeCe-TS group, dietary 0.3 mg/kg Se from SeCe with transport stress. The analyzed Se contents in Na_2_SeO_3_ and SeCe diets were 0.315 mg/kg and 0.318 mg/kg, respectively.

All birds were raised in wire-floored cages (120 × 120 × 60 cm^3^) in an environmentally controlled room with continuous light and had ad libitum access to feed and water. The ambient temperature was maintained at 36 °C at the start of experiment and was decreased as the birds progressed in age. The relative humidity was ranged from 45% to 55%.

On the morning of day 42, all birds were fasted 8 h overnight before. Then, the birds in transport stress groups were placed into 12 crates (120 × 60 × 30 cm^3^). All crates were randomly positioned in the same lorry (75 km/h, the inside temperature was maintained 25 ± 2 °C and the relative humidity was ranged at 42–44%). The transport time was from 07:00 to 10:00 a.m. The birds in the non-transport stress groups were kept in the original cages. Birds were not provided with feed and water during the transportation.

### 2.3. Sample Collection

Birds were weighed before and after transportation to calculate body weight loss. After transportation, one bird with near average body weight was chosen from each replicate and euthanized. Approximately 5 mL of blood from each bird was collected from the sub-wing vein. Plasma samples were extracted after centrifuging at 3500 rpm for 10 min and stored at −20 °C for biochemical parameters analysis. Heart, liver, spleen and bursa samples were collected and weighed to calculate the relative organ weight. Liver samples were quickly frozen in liquid nitrogen and transferred to a −80 °C freezer for anti-oxidative capacity analysis. The breast and thigh muscle were removed on the left side of the broilers for meat quality determination.

### 2.4. Plasma Biochemical Parameters

Plasma biochemical parameters including total protein (TP), glucose (GLU), triglyceride (TG), and blood urea nitrogen (BUN) levels. The activities of aspartate aminotransferase (AST) and alanine aminotransferase (ALT) were determined by automatic biochemistry analyzer (Hitachi 902 Automatic Analyzer, Hitachi, Tokyo, Japan) according to the manufacturer’s instructions.

### 2.5. Anti-Oxidative Capacity

Total anti-oxidative capacity (T-AOC), activities of glutathione peroxidases (GSH-Px), superoxide dismutases (SOD) and content of malondialdehyde (MDA) in liver were determined by spectrophotometric methods following the instructions of the commercial kits’ manufacturer (Nanjing Jiancheng Bioengineering Institute, Nanjing, China).

### 2.6. Meat Quality

Meat color (lightness, *L**; redness, *a**; and yellowness, *b**) was measured by a Chromameter (CR-410, Konica Minota, Tokyo, Japan) according to a previous study [14]. Drip loss was determined using the plastic bag method according to a previous study [15]. Briefly, about 30 g of muscle sample was weighed and set in a sealed plastic bag kept for 24 h at 4 °C. Then, the samples were taken out of the bags and dried. Drip loss was the loss of the sample weight to the initial sample weight. Cooking loss was determined according to a previous study [16]. About 30 g of muscle sample was weighed and set in a sealed plastic bag. Then, the samples were cooked at 80 °C for 20 min in a water bath. After cooking, the samples were cooled at room temperature and weighed. Cooking loss was the loss of the sample weight compared to the initial weight. Shear force was determined by a previous method [17]. Briefly, six technical replications of the cooked samples were taken parallel to muscle fibers to determine the maximal shear force (TA500 Lloyd Texture Analyzer fitted with a triangular Warner-Bratzler shear, Lloyd instruments, Bognor Regis, UK).

### 2.7. Statistical Analyses

All data were analyzed by ANOVA using the general linear model procedure of SAS 9.1 (SAS Institute, Cary, NC, USA) in a 2 × 2 factorial design with diet and stress as the main effects. Each replicate was the experimental unit for the body weight loss. Individual bird was the experimental unit for the other parameters. When the *p* value of the interaction between main effects was lower than 0.05, differences among the treatments were examined by Duncan’s multiple range test. Differences between means at *p* < 0.05 were considered to be significant. All data were expressed as means with their pooled SEM.

## 3. Results and Discussion

### 3.1. Body Weight Loss

Table 1 shows the effects of dietary SeCe supplementation on body weight loss of broilers under transport stress. After 42 days of the feeding trial, there was no significant difference in the body weight between the broilers fed Na_2_SeO_3_ diets and SeCe diets (*p* > 0.05). However, after a 3 h transportation, there was a significant interaction between diets and stress in the body weight loss of the broilers (*p* < 0.05). Dietary SeCe supplementation significantly alleviated transport-stress-induced body weight loss of the broilers compared with the broilers fed Na_2_SeO_3_ diets (*p* < 0.05). Typically, after 42 days implementation of intensive farming, broilers are transported from farms to slaughterhouses. The factors in the transportation process such as pre-slaughter feeding restriction, catching, crating and transportation, can all induce stress [18]. In addition, the transportation time and distance often lead to different levels of body weight loss [19]. Our study showed that the 3 h transportation treatment increased the body weight loss of broilers, which is consistent with previous studies [5,20]. Meanwhile, dietary SeCe supplementation alleviated the body weight loss of broilers under transportation stress. A previous study reported that the reasons of body weight loss during transport were emptying of gastrointestinal chyme, dehydration of broilers, and oxidative decomposition of body composition [21]. Because of the rich content of organic Se in SeCe, the new Se source SeCe may prevent the oxidative decomposition of body composition and enhance the efficiency of nutrient utilization by the birds.

### 3.2. Relative Organ Weight

Table 2 shows the effects of dietary SeCe supplementation on relative organ weight of broilers under transport stress. There was a significant interaction between diets and stress in the relative organ weight of liver (*p* < 0.05). Dietary SeCe supplementation significantly relieved transport stress-induced hepatomegaly of the broilers compared with broilers fed Na_2_SeO_3_ diets (*p* < 0.05). Additionally, the broilers fed SeCe diets had significantly increased relative organ weight of bursa compared with the broilers fed Na_2_SeO_3_ diets (*p* < 0.05). Relative organ weight reflects the healthy status of the tissue and organ, and abnormal values of relative organ weight are often accompanied by tissue or organ injury [22]. A previous study showed that transport stress led to liver injury induced by nutrient metabolic disorders, which is consistent with our study [23]. In the current study, dietary SeCe supplementation relieved transport stress-induced hepatomegaly of the broilers. There have been some reports showing that organic Se has the effect of liver protection [24]. Moreover, a previous study demonstrated that broilers fed organic Se had enhanced immune capacity, which is consistent with the increased relative organ weight of bursa of the broilers fed SeCe diets in the current study [25].

### 3.3. Plasma Biochemical Parameters

Table 3 shows the effects of dietary SeCe supplementation on plasma biochemical parameters of broilers under transport stress. There was a significant interaction between diets and stress in the concentrations of plasma TP and GLU (*p* < 0.05). Dietary SeCe supplementation significantly relieved transport stress-induced plasma TP and GLU level decline of the broilers compared with the broilers fed Na_2_SeO_3_ diets (*p* < 0.05). Additionally, there was a significant interaction between diet and stress in the concentrations of plasma BUN (*p* < 0.001). Dietary SeCe supplementation significantly alleviated transport stress-induced plasma BUN levels increase of the broilers compared with the broilers fed Na_2_SeO_3_ diets (*p* < 0.001). Furthermore, there was a significant interaction between diets and stress in the activities of plasma AST (*p* < 0.05) and ALT (*p* < 0.001). Dietary SeCe supplementation significantly alleviated transport stress-induced plasma AST (*p* < 0.05) and ALT (*p* < 0.001) activities increase of the broilers compared with the broilers fed Na_2_SeO_3_ diets. Protein and glucose are the main sources of energy for the body, and the liver is the organ for glycogen storage [26]. Under stress, hepatic glycogenolysis can be enhanced, and the glucose produced enters the blood to keep the normal concentration of glucose. When liver glycogen is depleted, body protein and lipid are catabolized to maintain the requirements of the body, which is reflected by an increased concentration of BUN [27]. In our study, dietary SeCe supplementation alleviated the decrease in plasma concentration of TP and GLU, and the increase of BUN concentration. These results demonstrate that dietary SeCe enhanced the energy and protein metabolism and utilization of the broilers under transport stress. Plasma AST and ALT activities are related to the health status of the liver [28]. In agreement with our results, a previous study showed that dietary organic Se supplementation can decrease the activities of AST and ALT in plasma [29]. The reason may be that SeCe has a better protecting effect on oxidative damage of the liver by regulating redox reactions. These results are in accordance with the results of the relative organ weight of the liver, which demonstrate that dietary SeCe enhanced the liver healthy status of the broilers under transport stress.

### 3.4. Anti-Oxidative Capacity

Table 4 shows the effects of dietary SeCe supplementation on liver anti-oxidative capacity of broilers under transport stress. The broilers fed SeCe diets had significantly increased liver T-AOC compared with the broilers fed Na_2_SeO_3_ diets (*p* < 0.05). There was a significant interaction between diets and stress in the activities of liver GSH-Px and SOD (*p* < 0.05). Dietary SeCe supplementation significantly relieved transport stress-induced liver GSH-Px and SOD activities decline of the broilers compared with the broilers fed Na_2_SeO_3_ diets (*p* < 0.05). Additionally, there was a significant interaction between diets and stress in the concentrations of liver MDA (*p* < 0.05). Dietary SeCe supplementation significantly alleviated transport stress-induced liver MDA levels increase of the broilers compared with the broilers fed Na_2_SeO_3_ diets (*p* < 0.05). The antioxidant system of poultry is consisted of non-enzymatic and enzymatic systems. T-AOC reflects the cumulative effect of all antioxidants in a non-enzymatic system [30]. GSH-Px and SOD are two main anti-oxidases, and their activities reflect the status of the enzymatic system [31]. MDA content reflects the degree of transport stress-induced lipid oxidation and the extent of cellular damage [32]. Our study showed dietary SeCe supplementation alleviated transport stress-induced liver anti-oxidases activities reduction, suggesting that SeCe improved the enzymatic system of transported broilers. With the rapid progress of broiler breeding, the anti-stress capacity of broilers is decreasing significantly, along with increased performance. The broilers are easily affected by external stressors such as transportation and high temperature leading to excessive production of ROS [33]. A previous study demonstrated that organic Se had higher anti-oxidative capacity compared with inorganic Se [34]. SeCe is a hyperaccumulating plant newly discovered in China and the Se mainly exists in SeCe in the organic form of SeCys_2_ and MeSeCys [12]. The better effect of dietary SeCe supplementation on anti-oxidative capacity of broilers under transport stress may be due to the form of organic Se occurring in SeCe.

### 3.5. Meat Quality

Table 5 shows the effects of dietary SeCe supplementation on meat quality of breast and thigh muscle in broilers under transport stress. There was a significant interaction between diets and stress in the redness (*a**) of breast muscle (*p* < 0.001). Dietary SeCe supplementation significantly improved the redness (*a**) of breast muscle compared with the broilers fed Na_2_SeO_3_ diets without transport stress (*p* < 0.001). Additionally, there was a significant interaction between diets and stress in the drip loss of breast muscle (*p* < 0.05). Dietary SeCe supplementation significantly alleviated transport stress-induced drip loss increase of the broilers compared with the broilers fed Na_2_SeO_3_ diets (*p* < 0.05). Furthermore, the broilers fed SeCe diets had decreased trend for shear force of breast muscle of the broilers compared with the broilers fed Na_2_SeO_3_ diets (*p* = 0.082). For thigh muscle, the broilers fed SeCe diets had significantly enhanced redness compared with the broilers fed Na_2_SeO_3_ diets (*p* < 0.05). Moreover, there was a significant interaction between diets and stress in cooking loss and shear force of thigh muscle (*p* < 0.05). Dietary SeCe supplementation significantly relieved transport stress-induced cooking loss and increased shear force of the broilers compared with the broilers fed Na_2_SeO_3_ diets (*p* < 0.05). Preslaughter transport is required in the broiler processing industry, and is detrimental to meat quality, leading to great economic loss to broiler production [35,36]. Consistently, the current study also showed that a 3 h transport treatment increased drip loss, cooking loss and shear force, indicating poorer meat quality. Drip loss and cooking loss are associated with the capacity of muscle proteins to hold water within the cells [37]. Shear force of the muscle reflects the tenderness associated with the mouthfeel of the meat [38]. Our study showed that dietary SeCe supplementation alleviated transport-stress-induced meat quality reduction of breast and thigh muscle in broilers. In agreement with our results, other studies have reported the better effect of SeCe on meat quality of broilers compared with Na_2_SeO_3_ [12,13]. The reason may be that Se increases oxidation resistance and prevents muscle protein being oxidized [39]. Moreover, the improved meat quality is in accordance with the results of enhanced performance and anti-oxidative capacity, which demonstrate that SeCe is a good feed additive to enhance meat quality of broilers under transport stress. From the above findings, SeCe can improve the health status and meat quality of the broilers under transport stress.

## 4. Conclusions

In conclusion, compared with Na_2_SeO_3_, dietary supplementation with 0.3 mg/kg Se from SeCe alleviates transport-stress-induced body weight loss, anti-oxidative capacity and meat quality impairment of broilers.

## Figures and Tables

**Table 1 animals-12-03193-t001:** Effects of dietary Se-enriched *Cardamine enshiensis* supplementation on body weight loss of broilers under transport stress ^1^.

Item	Non-Stress		Transport Stress		SEM	*p*-Value		
	Na_2_SeO_3_	SeCe	Na_2_SeO_3_	SeCe		Diets	Stress	Interaction
Initial body weight, g	2065	2135	2088	2107	30	0.459	0.968	0.899
Final body weight, g	2031	2103	1983	2032	32	0.325	0.551	0.251
Body weight loss, g	34 ^c^	32 ^c^	105 ^a^	75 ^b^	5	0.007	<0.001	<0.001

Na_2_SeO_3_, sodium selenite; SeCe, Se-enriched *Cardamine enshiensis*; SEM, standard error of the mean. ^a,b,c^ Within a row means followed by different letters are different at *p* < 0.05. ^1^ There were six replicates per treatment.

**Table 2 animals-12-03193-t002:** Effects of dietary Se-enriched *Cardamine enshiensis* supplementation on relative organ weight of broilers under transport stress ^1^.

Item	Non-Stress		Transport Stress		SEM	*p*-Value		
	Na_2_SeO_3_	SeCe	Na_2_SeO_3_	SeCe		Diets	Stress	Interaction
Heart, ×10^−3^	4.16	4.06	4.04	3.98	0.30	0.868	0.699	0.975
Liver, ×10^−3^	16.3 ^b^	16.5 ^b^	19.3 ^a^	16.3 ^b^	0.82	0.188	0.410	<0.001
Spleen, ×10^−3^	1.04	0.97	0.99	0.96	0.08	0.821	0.905	0.793
Bursa, ×10^−3^	1.86	2.28	1.78	2.35	0.16	0.039	0.752	0.609

Na_2_SeO_3_, sodium selenite; SeCe, Se-enriched *Cardamine enshiensis*; SEM, standard error of the mean. ^a,b^ Within a row means followed by different letters are different at *p* < 0.05. ^1^ There were 6 replicates per treatment.

**Table 3 animals-12-03193-t003:** Effects of dietary Se-enriched *Cardamine enshiensis* supplementation on plasma biochemical parameters of broilers under transport stress ^1^.

Item	Non-Stress		Transport Stress		SEM	*p*-Value		
	Na_2_SeO_3_	SeCe	Na_2_SeO_3_	SeCe		Diets	Stress	Interaction
TP, g/L	25.6 ^a^	25.8 ^a^	22.0 ^b^	24.8 ^a^	1.24	0.452	0.398	0.035
GLU, mmol/L	11.7 ^a^	11.9 ^a^	10.1 ^b^	12.2 ^a^	0.94	0.663	0.410	0.031
TG, mmol/L	2.85	2.65	2.68	2.59	0.15	0.358	0.774	0.814
BUN, mmol/L	0.45 ^b^	0.48 ^b^	0.72 ^a^	0.52 ^b^	0.05	0.696	0.099	<0.001
AST, U/L	215 ^b^	208 ^b^	288 ^a^	225 ^b^	19	0.089	0.035	0.017
ALT, U/L	4.36 ^b^	4.28 ^b^	5.27 ^a^	4.38 ^b^	0.22	0.469	0.280	<0.001

ALT, alanine aminotransferase; AST, aspartate aminotransferase; BUN, blood urea nitrogen; GLU, glucose; Na_2_SeO_3_, sodium seleniteix; SeCe, Se-enriched *Cardamine enshiensis*; SEM, standard error of the mean; TG, triglyceride; TP, total protein. ^a,b^ Within a row means followed by different letters are different at *p* < 0.05. ^1^ There were s replicates per treatment.

**Table 4 animals-12-03193-t004:** Effects of dietary Se-enriched *Cardamine enshiensis* supplementation on liver anti-oxidative capacity of broilers under transport stress ^1^.

Item	Non-Stress		Transport Stress		SEM	*p*-Value		
	Na_2_SeO_3_	SeCe	Na_2_SeO_3_	SeCe		Diets	Stress	Interaction
T-AOC, Mm	0.39	0.45	0.34	0.44	0.02	0.029	0.426	0.525
GSH-Px, U/mL	56.2 ^a^	54.6 ^a^	40.4 ^b^	60.3 ^a^	2.51	0.018	0.166	0.007
SOD, U/L	673 ^a^	658 ^a^	513 ^b^	754 ^a^	38	0.048	0.562	0.027
MDA, nmol/mL	1.49 ^b^	1.30 ^b^	1.90 ^a^	1.20 ^b^	0.11	0.030	0.302	0.019

GSH-Px, glutathione peroxidases; MDA, malondialdehyde; Na_2_SeO_3_, sodium selenite; SeCe, Se-enriched *Cardamine enshiensis*; SEM, standard error of the mean; SOD, superoxide dismutases; T-AOC, total anti-oxidative capacity. ^a,b^ Within a row means followed by different letters are different at *p* < 0.05. ^1^ There were 12 replicates per treatment.

**Table 5 animals-12-03193-t005:** Effects of dietary Se-enriched *Cardamine enshiensis* supplementation on meat quality of breast and thigh muscle in broilers under transport stress ^1^.

Item	Non-Stress		Transport Stress		SEM	*p*-Value		
	Na_2_SeO_3_	SeCe	Na_2_SeO_3_	SeCe		Diets	Stress	Interaction
Breast muscle								
Color								
*L**	66.9	69.3	65.8	65.4	2.85	0.774	0.412	0.680
*a**	11.6 ^b^	15.6 ^a^	11.8 ^b^	12.2 ^b^	0.90	0.128	0.620	<0.001
*b**	5.00	4.78	5.17	5.28	0.67	0.928	0.745	0.880
Drip loss, %	1.13 ^b^	0.80 ^b^	1.81 ^a^	1.10 ^b^	0.22	0.062	0.048	0.014
Cooking loss, %	30.4	28.2	35.5	35.9	1.20	0.741	<0.001	0.385
Shear force, N	27.0	24.8	27.4	25.0	1.01	0.082	0.710	0.608
Thigh muscle								
Color								
*L**	60.4	65.2	63.8	62.1	2.63	0.385	0.842	0.710
*a**	12.2	14.4	12.0	14.3	0.79	0.018	0.859	0.773
*b**	6.83	6.44	6.00	5.39	0.66	0.296	0.368	0.872
Drip loss, %	0.67	0.69	0.77	0.73	0.18	0.880	0.621	0.743
Cooking loss, %	33.7 ^ab^	26.0 ^c^	35.9 ^a^	31.1 ^b^	1.32	0.006	0.036	0.028
Shear force, N	26.9 ^b^	21.4 ^c^	28.8 ^a^	28.5 ^a^	0.80	0.023	0.008	<0.001

Na_2_SeO_3_, sodium selenite; SeCe, Se-enriched *Cardamine enshiensis*; SEM, standard error of the mean. ^a,b,c^ Within a row means followed by different letters are different at *p* < 0.05. ^1^ There were 12 replicates per treatment.

## Data Availability

The data presented in this study are available in the manuscript.

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
