# Peer review of "Dietary Se-Enriched Cardamine enshiensis Supplementation Alleviates Transport-Stress-Induced Body Weight Loss, Anti-Oxidative Capacity and Meat Quality Impairments of Broilers"

_animals, 2022, doi:10.3390/ani12223193_

Round 1
Reviewer 1 Report
Dear Editor;
I have reviewed the manuscript entitled "Dietary Se-enriched Cardamine enshiensis Supplementation Alleviates Transport-stress-induced Body Weight Loss, Anti-oxidative Capacity and Meat Quality Impairments of Broilers" (identified as animals-2042338). In the current study, authors aimed to compare the effects of two different sources of selenium (sodium selenite vs. Se-enriched Cardamine enshiensis) on body weight loss, anti-oxidative capacity and meat quality of broilers under transport stress using ROSS 308 broilers. The subject of the study falls within the scope of the journal. An adequate amount of information has been provided in the introduction, materials and methods, results and discussion. There are, however, some points that should be edited. The following is a detailed explanation of the points I have identified.
Major comments:
L59-60: In my opinion, this sentence hinders the fluency of the paragraph. Thus, I would suggest using the following sentence: “Therefore, the poultry industry is concerned about how to alleviate the transport stress.”. Furthermore, the last two sentences of the first paragraph appear to be independent of one another. I suggest connecting these two sentences by using a conjunction.
L159-160: When providing initial body weight results, a trial period is not necessary. However, a trial period can be given for final body weight. This sentence should be rewritten to include final body weight.
L179-181, L200-202, L234-237, L267-270, L303-305: Please rewrite it according to the spelling rules provided in the Word template. Furthermore, It is recommended to write explanations under each table of “Na2SeO3” and “SeCe”.
L274-276: Based on the data presented in Table 5, it is not possible to draw such an inference. It is clear from the table that the group with the highest redness value is the SeCe group, which is not subject to transport stress. In the study, Na2SeO3 addition to the diet did not affect the redness value in groups with and without transport stress. Furthermore, the SeCe group without transport stress was redder than the others.
L280-281: It would be more accurate if this sentence was rewritten. To discuss a trend, different doses should be used in the experimental groups. However, in the study described here, only one dose was used.
Minor comments:
L47-48: It is recommended that the keywords be sorted alphabetically.
L67: The genus name of the species given a binomial name in the text can be used directly for the next use. It is therefore unnecessary to use a separate abbreviation in parentheses for the next use. Therefore, “(C. enshiensis)” should be removed.
L102, 107, 116, 118, 138, 141, : Please give a space between "…"and "°C".
L115: Please use “rpm” instead of “rmp”.
L134: The abbreviations (L, a, b) for meat color parameters should be in italics.
L135, 136, L138-140, : The "previous studies" should be corrected since only one source is cited.
L142-144 : The "previous methods" should be corrected since only one method is used
L144-147: There should be a statement indicating how many technical replications are used to determine the shear force.
L167: Please use “broilers” instead of “birds”.
L172-174: It is necessary to provide more references due to the use of "some studies”.
L174-176: The source should be added if the sentence is being written in its current form. If it is believed to be caused by this, it should be rewritten.
L178: Please remove the dot after 1.
L191-192: It is necessary to provide more references due to the use of "some studies”.
L194-195: It is necessary to provide more references due to the use of "some reports”.
L195: Please use “organic” instead of “orgainic”.
L195-197: It is necessary to provide more references due to the use of "some studies”.
L210, 212: Please use “(p < 0.001)” instead of “(p < 0.05)”.
L213: “AST (p <0.05) and ALT (p < 0.001). “ instead of “AST and ALT (p < 0.05).”. The significance levels should be indicated as such in the following sentence.
L217-221: Try explaining it another way. You have used two consecutive sentences that begin with “When”. This disrupts the flow of the text.
L225-227: It is necessary to provide more references due to the use of "some studies”.
L228-231: It is important to cite the studies that are compatible with this study.
L243:Please check the extra space in this line.
L259-260: It is necessary to provide more references due to the use of "Previous studies”.
L276: Please use “(p < 0.001)” instead of “(p < 0.05)”.
L285: “increased shear force” instead of “shear force increase”.
L293, L295, L299: “SeCe” instead of “SeCv”.
L297-300: The source should be added if the sentence is being written in its current form. If it is believed to be caused by this, it should be rewritten.
L308-310: There can be a conclusion written emphasizing the study's most important findings.
Author Response
We have attached the point-by-point response to the reviewer's comments.

Reviewer 2 Report
Xu and colleagues investigated the effects of SeCe supplementation on transport-stress-induced body weight loss and meat quality impairments in broiler chickens. Overall, the manuscript is well-written and discussed. However, please include some information about how SeCe differs from other organic Se (for example cost-effectiveness and etc).
Line 32 & 96: change to “the 4 groups were as follows:”
How many birds were loaded into one crate? How long is the road transport distance and how many hours? Will a temperature of 25C stress the birds?
Line 293, 295, 299. correct the word “SeCv”. Check and correct all.
There is no information available about the diet and nutrients of broilers. What about the analyzed Se content in treatment diets that was studied?
The conclusions are too general. Please revise it.
Author Response

(The authors gave the same response as above.)
